# Intraoperative Autofluorescence and Indocyanine Green Angiography for the Detection and Preservation of Parathyroid Glands

**DOI:** 10.3390/jcm9030830

**Published:** 2020-03-18

**Authors:** Marco Stefano Demarchi, Wolfram Karenovics, Benoît Bédat, Frédéric Triponez

**Affiliations:** Department of Thoracic and Endocrine Surgery and Faculty of Medicine, University Hospitals of Geneva, 4 Rue Gabrielle Perret-Gentil, 1211 Geneva, Switzerland; marcostefano.demarchi@hcuge.ch (M.S.D.); wolfram.karenovics@hcuge.ch (W.K.); benoit.bedat@hcuge.ch (B.B.)

**Keywords:** near-infrared fluorescence imaging, parathyroid gland, thyroid gland, hypoparathyroidism, thyroid surgery

## Abstract

Fluorescence imaging is a well-known method for both the in vivo and in vitro identification of specific cells or tissues. This imaging tool is gaining importance in the intraoperative detection and preservation of parathyroid glands during endocrine surgery owing to the intrinsic properties of parathyroid tissue. The aim of this paper is to provide an overview of the basics of the technology, its history, and the recent surgical intraoperative applications of near-infrared imaging methods. Moreover, a literature review of the utilization of fluorescence devices in thyroid surgery suggests that the use of near-infrared imaging seems to be beneficial in reducing postoperative hypoparathyroidism, which is one of the most frequent complications of thyroid surgery.

## 1. Introduction

Hypoparathyroidism is one of the most frequent complications resulting from thyroid surgery, occurring in as many as 30% of patients [1,2]. During thyroid surgery, the parathyroid glands can be inadvertently excised or devascularized owing to their small size and the similarity of their color to fat tissue, even in the hands of the most experienced surgeons.

Thus, the localization and preservation of the parathyroid glands during thyroidectomy is of paramount importance for reducing the risk of postoperative hypocalcemia [3].

As imaging techniques have evolved, new devices have been introduced that allow surgeons to detect parathyroid glands more easily, thereby reducing morbidity due to thyroid surgery.

The purpose of this paper is to provide an overview of the basics of new technologies and the most recent applications of near-infrared imaging methods in thyroid surgery.

## 2. Discussion

The use of fluorescence imaging, a well-known method that is used in the biomedical sciences for the in vitro and in vivo visualization of cells and tissues, is relatively recent in surgery [4].

Fluorescence is the property of some substances and molecules to absorb external light at a given wavelength (i.e., excitation) and then to emit light at a different, longer wavelength with lower energy (i.e., emission).

With this model in mind, the principle of fluorescence imaging is simple: the tissue is (i) illuminated with a filtered light source at a specific wavelength, (ii) absorbs the excitation wavelengths, and (iii) emits a fluorescent band that can be captured by a camera that detects the longer emission wavelengths (which can be filtered from any reflected excitation light) [4,5].

There are two ways to highlight fluorescence in the area of interest in a given sample: (i) by using exogenously-administered contrast agents, which are either fixed in the target cells/tissues (the preferred method) or are flowed through the vessels, or (ii) via a label-free optical modality that relies only on the intrinsic properties of the tissue.

In endocrine surgery, temporary and definitive postoperative hypoparathyroidism are the most frequent complications of thyroidectomy [1,2,3,6] and result from the disruption of the parathyroid vasculature or the inadvertent excision of the parathyroid glands. Because the parathyroid glands are difficult to accurately identify by the naked eye, and their tiny vessels are even more difficult to distinguish and preserve, these complications can occur in the hands of the most experienced surgeons.

The need for a reliable technique to identify parathyroid glands intraoperatively has driven associated research since the first studies were conducted by Dudley in 1971 [7], who proposed methylene blue as an exogenous contrast agent. However, because no consistent benefits were identified and because of the potential toxicity associated with the use of methylene blue (e.g., mainly the risk of serious neurological adverse effects), the literature has since discouraged the routine use of this contrast agent [8,9,10].

While authors such as Van der Vorst et al. (2014) [11] and Prosst et al. (2014) [12] have explored new solutions to this issue, including, respectively, the use of near-infrared fluorescence imaging after the intravenous application of smaller doses of methylene blue and aminolevulinic acid (ALA) as a contrast agent, these techniques did not gain popularity because of difficulties encountered during their application in clinical practice and low detection rates of the parathyroid gland (approximately 45%) [12,13,14].

The real revolution in the intraoperative identification of parathyroid glands dates back to 2008, when a poster presentation at the American Association of Endocrine Surgeons in Monterey, CA (USA) highlighted the preliminary results of a special optical characteristic of the parathyroid glands: autofluorescence. However, it was not until 2011 that this same team of biomedical engineers, who worked at Vanderbilt University (Nashville, TN, USA), first published on the autofluorescent properties of parathyroid tissue in the near-infrared spectrum [15].

Parathyroid tissue exhibits a unique autofluorescence signature when excited at the near-infrared (NIR) wavelength of 785 nm, reemitting at a wavelength between at 820 to 830 nm, with an intensity that is two to 11 times greater than that of the surrounding tissues (e.g., thyroid tissues—Figure 1), thereby allowing for the improved detection and the precise localization of the parathyroid glands.

The endogenous fluorophore that is responsible for this particular optical effect in the parathyroid gland remains unknown, although some authors believe that the molecule could be either a calcium-sensing or vitamin D receptor [16,17].

Since the publication of this discovery, many studies have reported excellent parathyroid detection rates, as described by McWade et al. in 2014 [16,17,18], with a specificity of more than 80% [19].

However, one limitation of this technique is that the difference between thyroid and parathyroid tissues can be difficult to distinguish owing to the similarity of the autofluorescence intensities of the two tissues in some cases (i.e., the thyroid can be more autofluorescent in some thyroid states, such as thyroiditis, which reduces the contrast between thyroid and parathyroid tissues).

Moreover, this intraoperative imaging technique results in false positives, where brown fat, colloidal nodules, or metastatic lymph nodes may exhibit an autofluorescence that overlaps that of the parathyroid tissue [20,21].

Subsequently, the industry began to develop devices based on autofluorescence in the near-infrared spectrum. Currently, two of these devices are approved by the FDA for performing the real-time identification of parathyroid tissues during surgery: Fluobeam^®^ (Fluoptics©, Grenoble, France) and the Parathyroid Detection PTeye System (AIBiomed Inc., Santa Barbara, CA, USA).

The Fluobeam^®^ is an optical system that reproduces a real-time grayscale image with the detection and enhancement of autofluorescent tissues (Figure 2). From a practical point of view, this device consists of a laser emitter associated with a camera that is able to detect the autofluorescence emitted from the explored tissue, with the resulting black and white image then projected on a screen. During the intervention, this device, which includes a dedicated sterile cover, is used directly by the surgeon to explore the surgical field.

In contrast, the PTeye System is a sterile probe that analyzes the optical properties of the tissues at the tip of the probe and provides a distinct audio and visual signal when the probe is touching parathyroid tissue.

The most important and recent studies, which are summarized in Table 1, mainly demonstrate that the use of near-infrared autofluorescence (NIRAF) can be beneficial during thyroid surgery, in addition to facilitating the identification of parathyroid glands, reducing the rate of postoperative hypoparathyroidism, and therefore improving surgical outcomes.

The NIRAF imaging of parathyroid tissue also demonstrates another particularly useful feature of parathyroid tissue that facilitates the identification of a diseased parathyroid gland. In general, parathyroid adenomas exhibit a less intense and more heterogeneous autofluorescence pattern in comparison to that of normal parathyroid glands [28] (Figure 3, Figure 4, Figure 5 and Figure 6). Thus, this imaging technique can also help to differentiate a normofunctioning parathyroid gland from a diseased parathyroid gland, based on the gland’s autofluorescence patterns.

Despite the benefits, autofluorescence is neither able to analyze the perfusion status of the parathyroid tissues nor the vitality of parathyroid glands. The properties of autofluorescence are also preserved after gland resection, with the fluorophore known to be resistant to heat, freezing, and formalin fixation [17,20]. As previously mentioned, the vitality of the parathyroid gland is of paramount importance in thyroid surgery when aiming to reduce postoperative hypoparathyroidism, which remains the most common complication of thyroidectomy, occurring in as many as 30% of patients [1,6].

The use of indocyanine green (ICG) to enhance fluorescence imaging enables the real-time assessment and direct imaging of tissue perfusion and vascularization. ICG is the only clinically approved NIR fluorescent dye, with approval first received for clinical use in 1956. Because this dye fluoresces a characteristic wavelength in the range of 830–845 nm when excited in plasma with 750 to 800 nm light, indocyanine green angiography has been used for decades. Its earliest applications, dating to the 1970s, were in ophthalmology, and included retinal angiography, although its initial use was in the detection of macular degeneration [4,5]. In the last few years, this technique has demonstrated its utility in the real-time assessment of intestinal microvascularization when evaluating intestinal anastomoses during colorectal surgery [29,30,31] or various other procedures including intraoperative angiography in reconstructive surgery, cholangiography, and lymph node mapping and dissection.

In addition, ICG has most recently emerged as a technique for assessing the vascularization of the parathyroid glands, which seems to be closely correlated with parathyroid function, after thyroid resection [22,32,33,34,35,36]. From a technical point of view, the protocols and doses of dye used for ICG angiography differ from center to center, with the best dosage still under debate. Until the study by Vidal et al. [32], ICG had been prepared by mixing ICG powder with 10 mL sterile water, followed by the intravenous injection of 3.5 mL of solution. This injection could be repeated until a maximum dose of 5 mg per kg. The catheter is then rapidly purged after each injection to acquire a rapid image gain. After approximately 30 s to 2 min, images are acquired with a NIRAF camera, showing the distribution of ICG in the explored tissue.

Today, the parathyroid angiography technique is feasible and is also gaining interest in mini invasive and robotic surgery applications, such as the Transoral Endoscopic Thyroidectomy Vestibular Approach (TOETVA) or the robotic bilateral axillo-breast approach (BABA), although at the time of authoring this manuscript, no articles were found on this subject.

A review of the literature on PG viability analyzed with intraoperative ICG angiography shows the articles presented in Table 2.

The results of these studies are mainly concordant with regard to the concept that patients who have retained at least one well-vascularized parathyroid gland after total thyroidectomy, as demonstrated by ICG angiography (Figure 7), also exhibited normal PTH levels during the first postoperative day, thereby excluding postoperative hypoparathyroidism with a 100% positive predictive value [33,37,39,40,41].

In contrast, some studies report fewer positive results regarding the prediction of parathyroid function using ICG angiography [34,38], although this difference could be related to the subjective visual interpretation of gray scale imaging (i.e., ICG score) [32] during angiography in the absence of a standardized numerical criterion (Table 3).

However, all published studies reported the recovery of parathyroid function over the long term, suggesting that parathyroid function also resumes in patients with a “moderately well vascularized parathyroid gland”.

Lastly, although ICG angiography is still “passive” because it only allows for the evaluation of parathyroid glands after surgical resection, nonetheless research that focuses on making these new imaging tools “active” in order to reduce the extent of thyroid resection, and thereby decreasing (ideally to zero) the rates of post thyroidectomy hypoparathyroidism, is very promising [36].

## 3. Conclusions

NIRAF imaging, which provides valuable spatial and anatomical information, allows for the real-time detection of parathyroid glands. This technique appears to be a new and valuable intraoperative tool for identifying both healthy and diseased PGs, which ultimately reduces postoperative hypoparathyroidism and ensures better patient outcomes for thyroid surgery.

## Figures and Tables

**Figure 1 jcm-09-00830-f001:**
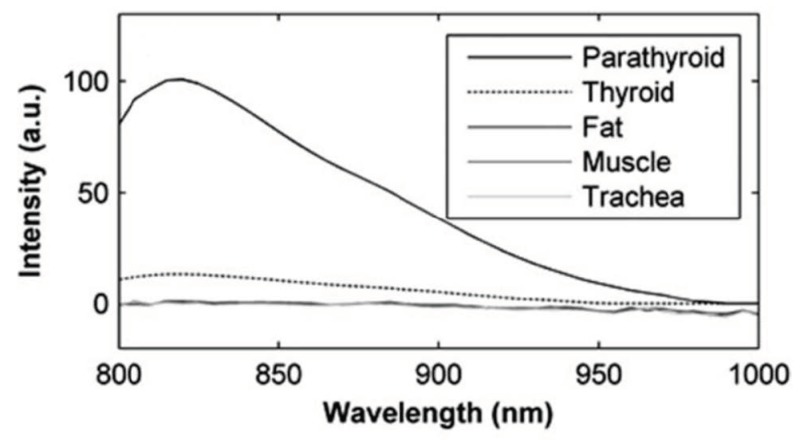
Comparison of the autofluorescence intensities of different tissues (note that the traces for fat, muscle, and trachea are not discernable because they are superimposed) [15].

**Figure 2 jcm-09-00830-f002:**
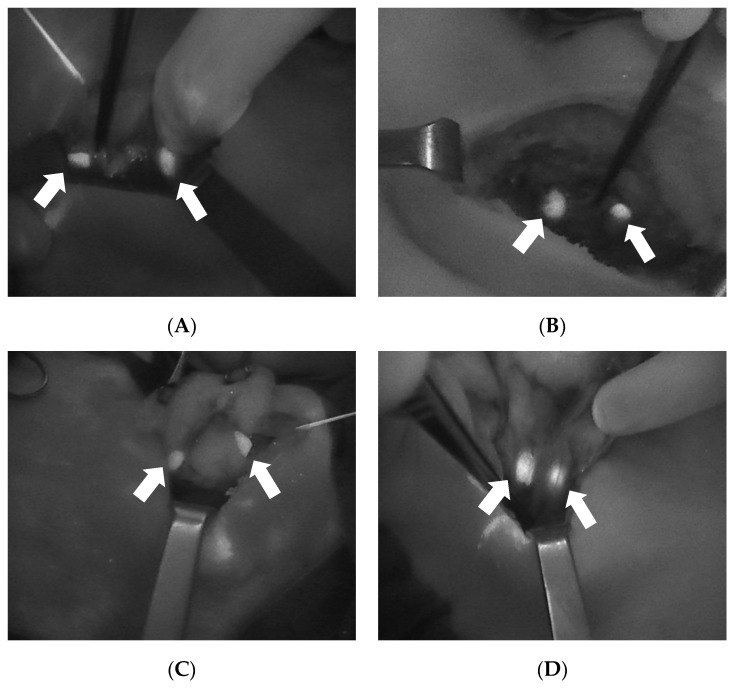
Four intraoperative images showing the autofluorescence of the parathyroid glands (arrows) as detected with Fluobeam LX^®^ (Fluoptics©, Grenoble, France). (**A**): Two PGs after right lobectomy. (**B**): Two PGs after left thyroid lobectomy. (**C**): Two PGs after superior pole dissection and the medialization of the left thyroid lobe. (**D**): Two PGs after superior pole dissection and the medialization of the right thyroid lobe. PGs, parathyroid glands.

**Figure 3 jcm-09-00830-f003:**
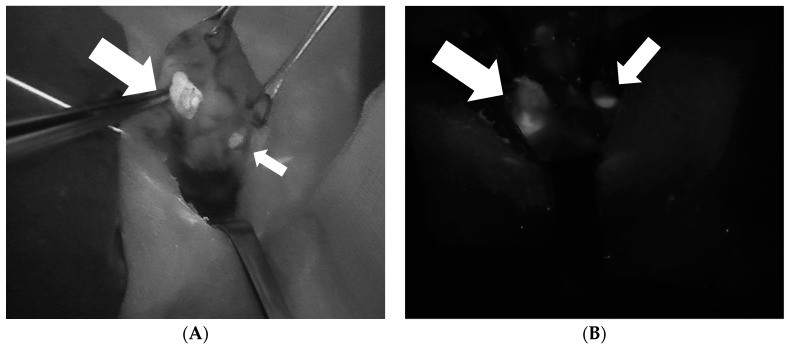
Two intraoperative images of parathyroid adenomas (larger arrow) demonstrating the heterogeneous fluorescence pattern that differentiates abnormal and normal parathyroid glands (smaller arrow). (**A**): Superior right parathyroid adenoma. (**B**): Inferior left parathyroid adenoma.

**Figure 4 jcm-09-00830-f004:**
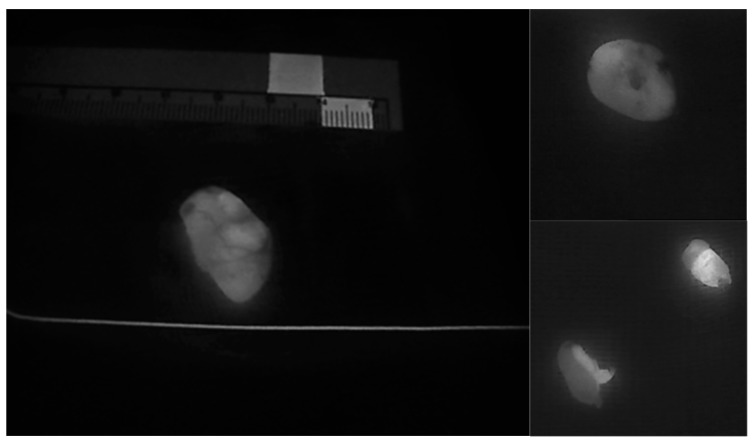
Heterogeneous autofluorescence patterns in parathyroid adenomas.

**Figure 5 jcm-09-00830-f005:**
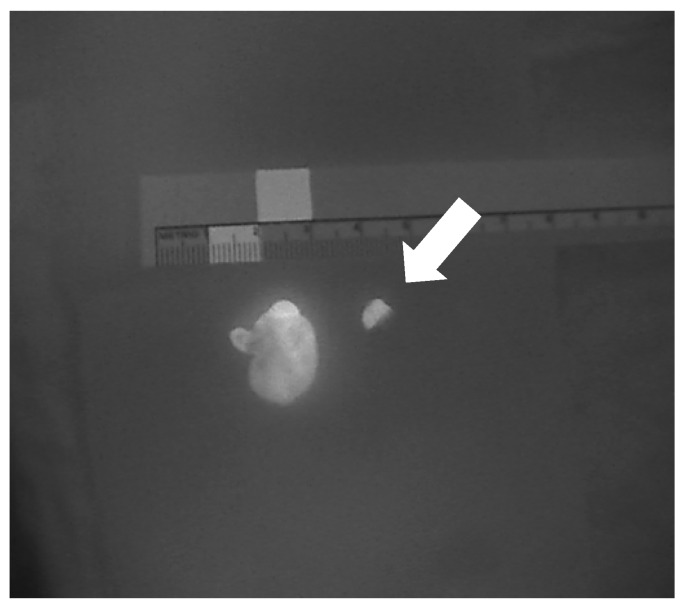
A resected parathyroid adenoma in comparison to a normal parathyroid gland (arrow).

**Figure 6 jcm-09-00830-f006:**
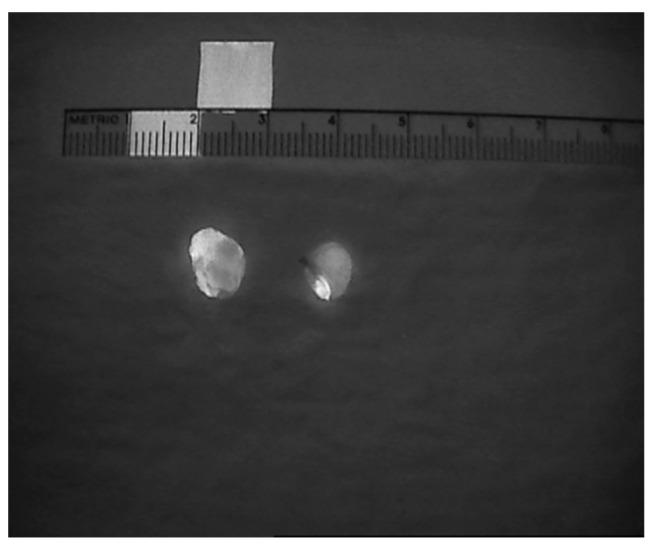
Heterogeneous fluorescent patterns in two resected parathyroid adenomas.

**Figure 7 jcm-09-00830-f007:**
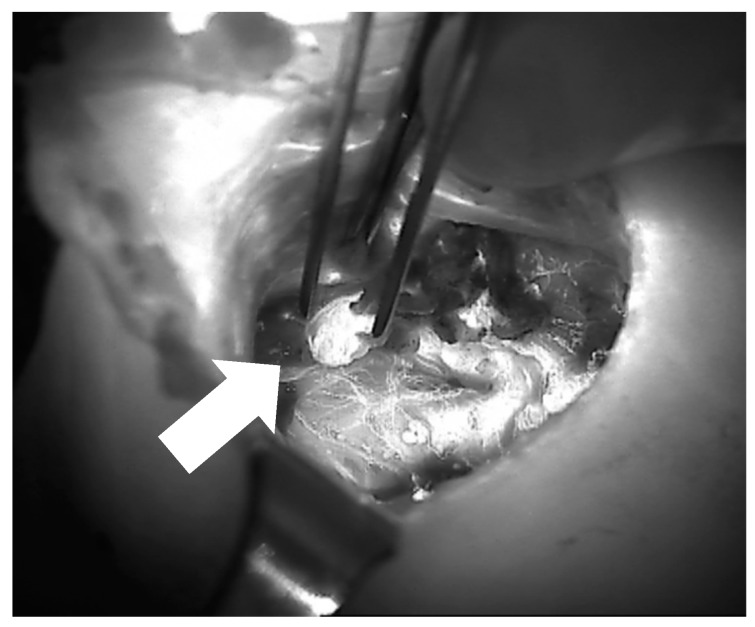
Image of parathyroid ICG angiography (of the right superior parathyroid after thyroidectomy) showing a well-vascularized parathyroid gland between the forceps (arrow).

**Table 1 jcm-09-00830-t001:** Recent studies that evaluate the benefits of near-infrared (NIR) imaging in thyroid surgery.

Author	Year	Type	Reduction of Hypoparathyroidism	Reduction in Inadvertent Resection of PG	Number of Patients	Increase in PG Detection	Key Notes
Benmiloud et al. [22]	2019	randomized controlled clinical trial	Yes, from 22 to 9% —PTH (*p* = 0.007)	Yes, from 12 to 3% (*p* = 0.006)	241	N/A	NIRAF helped reduce the rates of temporary postoperative hypocalcemia, parathyroid autotransplantation, and inadvertent parathyroid resection
Dip et al. [23]	2019	randomized controlled clinical trial	Yes, from 16.5 to 8.2% —hypocalcemia rates (*p* < 0.103)	N/A	170	Yes, from 2.6 to 3.5 (*p* < 0.001)	NIRAF increased and allowed the earlier intraoperative identification of parathyroid glands compared to white light alone
DiMarco et al. [24]	2019	randomized controlled clinical trial	No	No	269	N/A	NIRAF did not minimize inadvertent parathyroidectomy or postsurgical hypocalcemia in this study
Kahramangil et al. [25]	2018	retrospective Institutional review	N/A	N/A	210	Yes	NIFI facilitated PG identification by detecting gland AF, before conventional recognition by the surgeon
Falco et al. [26]	2017	retrospective review	N/A	N/A	74	Yes from 2.5 to 3.7 (*p* < 0.001)	NIRAF significantly increased the number of PGs identified during thyroid and parathyroid surgery
Benmiloud et al. [27]	2018	before and after controlled study	Yes, from 20.9 to 5.2% —hypocalcemia rates (*p* < 0.001)	Yes, from 7.2 to 1.1%	513	N/A	NIRAF use during total thyroidectomy significantly reduced postoperative hypocalcemia, improved parathyroid identification, and reduced rates of autotransplantation

NIR: near-infrared; NIRAF: near-infrared autofluorescence; AF, auto fluorescence; NIFI: near-infrared imaging; PTH, parathyroid hormone; PG, parathyroid gland.

**Table 2 jcm-09-00830-t002:** Recent studies that evaluate the benefits of indocyanine green angiography (ICGA) in thyroid surgery.

Author	Year	Nb pt	Intervention	Key Points	Conclusions
Lang et al. [37]	2017	70	TT	Greatest ICG correlated with postoperative normal PTH.	ICGA is a promising operative adjunct in determining residual parathyroid gland function and predicting postoperative hypocalcemia risk after total thyroidectomy.
Rudin et al. [34]	2019	210	TT	At least two vascularized glands on ICGA may predict postoperative parathyroid gland function.	ICGA is a novel technique that may improve the assessment of parathyroid gland blood supply compared to visual inspection.
Razavi et al. [38]	2019	111	TT	No significant difference in mean PTH changes at the end of surgery, symptomatic hypocalcemia, or length of stay between surgeries performed with and without ICG.	Low-flow ICG patterns are not associated with postoperative PTH changes or transient hypocalcemia and may lead to unnecessary parathyroid auto transplantation.
Gálvez-Pastor et al. [39]	2019	39	TT	Patients with postoperative hypocalcemia had a lower 4-ICG score. The 4-ICG score showed good discrimination in terms of predicting postoperative hypocalcemia.	The 4-ICG score predicts postoperative hypocalcemia and correlates well with postoperative parathyroid function in patients undergoing total thyroidectomy for multinodular goiter.
Jin et al. [40]	2019	26	TT	In patients with at least one parathyroid gland with an ICG score of 2 postoperative PTH levels were in the normal range.	Fluorescence imaging system applied with indocyanine green is a safe, easy, and effective method for protecting the parathyroid glnd and predicting postoperative hypoparathyroidism.
Vidal Fortuny et al. [33]	2018	196	TT	One well perfused parathyroid gland using ICGA is a reliable way of predicting the absence of postoperative hypoparathyroidism.	ICGA reliably predicts the vascularization of the parathyroid glands and obviates the need for the postoperative measurement of calcium and PTH.
Alesina et al. [41]	2018	5	TT/PHPT	Postoperative day 1 PTH was normal.	Superiority of combined AF/ICG vs. simple visualization to reduce the rate of postop HPOPT has not been demonstrated.

Nb pt, number of patients; ICG, indocyanine green; TT, total thyroidectomy; PTH, parathyroid hormone; PHPT, primary hyperparathyroidism; AF, auto fluorescence; HPOPT, hypoparathyroidism.

**Table 3 jcm-09-00830-t003:** Evaluation of parathyroid gland vascularization according to ICG score.

ICG score 0	Devascularized parathyroid gland
ICG score 1	Moderately well-vascularized parathyroid gland
ICG score 2	Well-vascularized parathyroid gland

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
