# Peer review of "Intraoperative Autofluorescence and Indocyanine Green Angiography for the Detection and Preservation of Parathyroid Glands"

_jcm, 2020, doi:10.3390/jcm9030830_

Round 1

Reviewer 1 Report

Demarchi et al review fluorescence methods in intraoperative imaging of parathyroid gland. The paper is not written in an academical style and does read as if it has not been proof-read. Reads like a piece of a google text-to-speech that the Authors recorded whilst driving to work. Needs re-writing and possibly re-thinking.

Title: unreadable, needs to be cut by 60% to sound exciting to the audience.

Some in-text citations bear a double formatting (authors, year + numbers)

Page 1: whereas 'Discussion' is a typical component of an experimental paper, I am not sure how sensible it is to start the review with that.
're-emission' can be substituted by just 'emission'

Page 2:

1 para: this is not an assumption but a well-described phenomenon

'filtered light source' = source of monochromated light

Fig.1: no reference in the text, poor quality copy-paste. Is in needed?

Why would the receptor contain a fluorophore?

Page 3:

4th para: what was the logics behind creating these devices

Fig 2: what are we looking at? Arrows, scale bar...

 Page 4:

Not sure about the message of Table 1. Lots pf 'nos' or 'NAs' and then the 'Use of NIRAF' column with 'yeses'. Increase in PG detection by Falco et al is opposite to that by Dip et al

Page 5:

Where do Figs 3-6 come from? Is this the Authors' own data - needs to be marked, respectively.

Can we have some critical analysis of the background/reasons for the autofluorescence of the adenomas?

Page 6:

2nd para: 'this technique' - which technique?

I am not sure if we can immediately take the ICG as a readout of viability/vitality. This needs discussion.

Author Response

Point 1: 

Title: unreadable, needs to be cut by 60% to sound exciting to the audience.

Response 1: We have revised and shortened the title.

Point 2: 

Some in-text citations bear a double formatting (authors, year + numbers)

Response 2: We have revised the formatting of our in-text citations.

Point 3: 

Page 1: whereas 'Discussion' is a typical component of an experimental paper, I am not sure how sensible it is to start the review with that.
're-emission' can be substituted by just 'emission'

Response 3: The purpose of the discussion is to interpret and describe the significance of our findings in light of what was already known about the research problem being investigated, thus we find it appropriate to call this section “discussion”. Moreover, we have followed the guidelines of MDPI on the structure of the article.

We revised “re-emission” to “emission”.

Point 4: 

Page 2:

1 para: this is not an assumption but a well-described phenomenon

This has been revised in the text.

'filtered light source' = source of monochromated light

We find it more appropriate to use the term “filtered light source” because the light is passed through a filter.

Fig.1: no reference in the text, poor quality copy-paste. Is in needed?

We added the reference in the text and changed the quality of the image.

Why would the receptor contain a fluorophore?

It is believed that the receptor contains a fluorophore because the receptor exhibits fluorescence but the fluorophore remains unknown.

Page 3:

4th para: what was the logics behind creating these devices

Our goal was to detect autofluorescence.

Fig 2: what are we looking at? Arrows, scale bar...

We have added arrows to the image.

 Page 4:

Not sure about the message of Table 1. Lots pf 'nos' or 'NAs' and then the 'Use of NIRAF' column with 'yeses'. Increase in PG detection by Falco et al is opposite to that by Dip et al

Because the various studies also entailed different designs, we found it necessary to use “NA” if the specific topic was not covered in the article.

We have corrected the detection rate of Falco and eliminated the column NIRAF.

Page 5:

Where do Figs 3-6 come from? Is this the Authors' own data - needs to be marked, respectively.

Can we have some critical analysis of the background/reasons for the autofluorescence of the adenomas?

The figures are our own data.

The reason for the different fluorescence patterns is not known, but the tissue is still fluorescent because it is a (pathological) parathyroid tissue.

Page 6:

2nd para: 'this technique' - which technique?

The paragraph is linked to the previous paragraph but was separated by images. We have linked the two paragraphs better (page 5 lines 1-6).

I am not sure if we can immediately take the ICG as a readout of viability/vitality. This needs discussion.

This finding was demonstrated by:

Vidal Fortuny, J. et al. Randomized clinical trial of intraoperative parathyroid gland angiography with indocyanine green fluorescence predicting parathyroid function after thyroid surgery. Br. J. Surg. 105, 350–357 (2018)

which demonstrates the correlation between ICG score and parathyroid function.

Moreover, we have added a table with articles demonstrating the relationship among ICG angiography and perfusion.

Reviewer 2 Report

The paper is well written. Nowadays ICG is useful in many branches of surgery, especially in parathyroid surgery and it is clearly explained.

Good job

Author Response

Thank you for your suggested revisions. We appreciated your comments.

We have made some minor changes to the article following the comments of the other reviewers.

Reviewer 3 Report

Thank-you for your excellent work. 

I enjoyed reviewing your review, especially in regards to NIRAF. However, the review seems to be unbalanced as the content regarding IGC fluorescence is too short and simple.

As a review article, I think it is appropriate for the authors to elaborate on the history of ICG, just as they did for auto-fluorescence so that the reader could grasp some perspective as to how the technology came to be, how it is being applied currently in endocrine surgery today and how it will be develoed in the future. There are some recently published review article (with which I personally do not have any conflict of interest) that the authors may refer to.

Moreover, just as the authors did with table 1, I think they should make a table for recent publications regarding ICG and their clinical results. 

Author Response

Dear Reviewer,

Thank you for your suggestions and review of our paper.

As suggested, we have expanded the section on ICG angiography in thyroid surgery and added a table with the most recent and important articles published on the subject.

We thank you kindly for your excellent suggestions, of which we completely approve.

Reviewer 4 Report

The authors reviewed intraoperative parathyroid detection technique during thyroid surgery, including autofluroscence and IGC angiography. This is one of the interesting topic to endocrine surgeons. This NIR imaging or other parathyroid imaging techniques were reported from many centers.

Assume that this article is kind of review article, readers may want to know the surgical instructions or ways to use the autofluroscence technique in thyroid surgery. I suggest that authos can be summarize the surgical techniques to parathyroid imaging using autofluroscence technique, prior to the discussion section, as the adding the method section.

Also, there's many reports about the endoscopic and robot assisted near-infrared imaging technique. If the authors can add up these parathyroid imaging in endoscopic / robotic surgery section, value of this article will be much improved.

Author Response

Dear Reviewer,

Thank you for your suggestions and review of our paper.

As suggested, we added a section that summarizes the surgical techniques using Fluobeam.

We also added a section that mentions the use of endoscopic NIRAF devices.

Round 2

Reviewer 1 Report

The readability of the paper has improved following the revisions.

Reviewer 3 Report

Excellent work on further explaining ICG application. The article seems much more thorough and balanced.

Thank-you very much for your informative review

Reviewer 4 Report

The authors answered my questions well. This review article will be helpful to many surgeons who are interested in image guided surgery.